# Test@Work Texts: Mobile Phone Messaging to Increase Awareness of HIV and HIV Testing in UK Construction Employees during the COVID-19 Pandemic

**DOI:** 10.3390/ijerph17217819

**Published:** 2020-10-26

**Authors:** Matthew Middleton, Sarah Somerset, Catrin Evans, Holly Blake

**Affiliations:** 1School of Medicine, University of Nottingham, Nottingham NG7 2UH, UK; mzymm16@nottingham.ac.uk; 2School of Health Sciences, University of Nottingham, Nottingham NG7 2HA, UK; sarah.somerset@nottingham.ac.uk (S.S.); catrin.evans@nottingham.ac.uk (C.E.); 3NIHR Nottingham Biomedical Research Centre, Nottingham NG7 2UH, UK

**Keywords:** HIV, workplace intervention, SMS, HIV testing, construction, mobile phone, COVID-19, health promotion, text messagingealth promotion kd

## Abstract

*Background:* HIV poses a threat to global health. With effective treatment options available, education and testing strategies are essential in preventing transmission. Text messaging is an effective tool for health promotion and can be used to target higher risk populations. This study reports on the design, delivery and testing of a mobile text messaging SMS intervention for HIV prevention and awareness, aimed at adults in the construction industry and delivered during the COVID-19 pandemic. *Method:* Participants were recruited at Test@Work workplace health promotion events (21 sites, *n* = 464 employees), including health checks with HIV testing. Message development was based on a participatory design and included a focus group (*n* = 9) and message fidelity testing (*n* = 291) with assessment of intervention uptake, reach, acceptability, and engagement. Barriers to HIV testing were identified and mapped to the COM-B behavioural model. 23 one-way push SMS messages (19 included short web links) were generated and fidelity tested, then sent via automated SMS to two employee cohorts over a 10-week period during the COVID-19 pandemic. Engagement metrics measured were: opt-outs, SMS delivered/read, number of clicks per web link, four two-way pull messages exploring repeat HIV testing, learning new information, perceived usefulness and behaviour change. *Results:* 291 people participated (68.3% of eligible attendees). A total of 7726 messages were sent between March and June 2020, with 91.6% successfully delivered (100% read). 12.4% of participants opted out over 10 weeks. Of delivered messages, links were clicked an average of 14.4% times, max 24.1% for HIV related links. The number of clicks on web links declined over time (*r* = −6.24, *p* = 0.01). Response rate for two-way pull messages was 13.7% of participants. Since the workplace HIV test offer at recruitment, 21.6% reported having taken a further HIV test. Qualitative replies indicated behavioural influence of messaging on exercise, lifestyle behaviours and intention to HIV test. *Conclusions*: SMS messaging for HIV prevention and awareness is acceptable to adults in the construction industry, has high uptake, low attrition and good engagement with message content, when delivered during a global pandemic. Data collection methods may need refinement for audience, and effect of COVID-19 on results is yet to be understood.

## 1. Introduction

HIV continues to pose a threat to global health. In 2019 there were 38 million people worldwide living with HIV, of which 25.5 million (67%) were receiving effective antiretroviral treatment [1]. The number of people living with HIV in the UK in 2018 was 77,305 [2]. There has been great progress made in HIV treatment in the last decade and it is now considered a manageable long term condition [3]. In the UK, new infection rates have fallen over the last five years [4], 98% of those infected are on antiretroviral treatment and 97% of these have an undetectable viral load which prevents onward transmission [5]. One of the highest risk factors for onwards transmission is late diagnosis, an important predictor of morbidity and mortality in HIV infection [6]. Those diagnosed late have a 11-fold risk of death compared with those diagnosed promptly and also have a higher risk of passing HIV to a new sexual partner [7]. Similarly, people who were newly diagnosed have been shown to be living with the infection for around three to five years before diagnosis [8] and it is estimated that 7500 (5400–11,500) people are living in the UK with undiagnosed HIV [4]. In the UK, HIV testing is mainly performed in sexual health service settings with 1.9 million attendees in 2018, yet just over 1.1 million (57.9%) had an HIV test upon consultation. Hence, a key component of UK HIV policy is to increase HIV testing both in clinical and non-clinical settings [9].

One way to increase uptake of HIV testing is to raise awareness of HIV and to promote testing outside of clinical services, such as in the workplace. Since 75.6% of the UK population are in employment [10]. Workplace settings offer a potentially important route for HIV-related health promotion initiatives. There are many potential strategies for workplace health promotion, including digital approaches which offer flexibility and can have wide reach with relatively low-cost implications [11]. UK mobile phone ownership is high with 96% of the population having access to a mobile phone [12]. Mobile phone based text messaging is acceptable to younger [13,14] and older populations [15,16,17]. Text-based health promotion has been shown to be effective in improving a range of health behaviours in various settings and contexts [11,18] and text messaging interventions can have ‘read rates’ as high as 98% compared with other forms such as email alone and are widely accessible, since they do not require a data plan or Wi-Fi signal to work [19].

Text messaging is commonplace in clinical settings and is widely used in the UK National Health Service (NHS) to encourage adherence to medication (e.g., asthma medication [20]), treatment regimens and appointments [21] and shown to be effective for other health promotion initiatives (e.g., antiretroviral therapy [11], obesity [22,23], type 1 diabetes [11,24]), to advocate health protection behaviours (e.g., birth control and sexual behaviours [11]; sexually transmitted infection (STI) prevention [13,25]), and to notify patients of health testing results (e.g., in sexual health services: chlamydia and gonorrhoea test results [26]). However, sexual health interventions rarely include working-age adults; they are often targeted to students and younger people, likely because they are seen to be vulnerable, lacking the associated health education and more likely to engage in high risk behaviours for contracting sexually transmitted infections [27].

Yet evidence suggests that working age adults are also at risk of sexually transmitted infections, and that this risk is higher in certain occupational groups, such as male, migrant, and low-waged workers (for example, those in the construction and agriculture industries) [28]. International studies have found a high prevalence of risky sexual behaviour amongst construction workers (44%) [29], such as sex with non-regular partners or while under the influence of alcohol, coupled with poor knowledge about sexual health and low rates of condom use [30]. Unhealthy lifestyle behaviours are common in the construction industry (e.g., poor diet, smoking, problem alcohol consumption, substance abuse) [28,31,32], with increased risk of mortality from drug use and accidental drug poisoning in construction workers (e.g., male painters and decorators, bricklayers and masons, plasterers, and roofers and glaziers) [33]. The prevalence of cardiovascular risk and mental ill-health is also increasing in construction workers [34,35,36]. HIV infections in adults have long been associated with health inequality, often correlating with the aforementioned risk behaviours [37,38]. Construction workers may be at increased risk for HIV due to occupational hazards [39,40], and migrant workers in the industry may be particularly vulnerable if they are from countries of high HIV prevalence [41,42]. As such, adults working in construction and associated industries are becoming a focus for workplace health interventions including HIV testing in an effort to improve detection, normalise HIV testing and remove stigma associated with sexual health and HIV [43].

In the workplace setting, mobile phone text messaging has been used to encourage employee adherence to safety initiatives and guidelines (e.g., safety queries [44]; reporting accidents [45]) and to promote a range of health behaviours to employees (e.g., weight control [46]; smoking cessation [15]; physical activity [16,47,48]). Workplace text messaging interventions have included those targeted specifically at low-income workers (obesity and weight loss [22,23]) as well as workers with known health risks (e.g., construction workers and sun safety [49]). Despite workplaces being a useful platform for health promotion, employers do not tend to include health testing in their health promotion offers, and when they do, this rarely, if ever, includes sexual health or HIV testing and awareness [50,51]. Only a small number of UK studies have reported on text messaging interventions aimed at raising awareness of HIV and increasing uptake of HIV testing (migrant African communities: Evans et al. [25]) and only one of these was delivered through the workplace setting [52].

To our knowledge, this is the first study to offer text messaging to raise awareness of HIV and testing among employees in the construction industry, an occupational group that may be at increased risk for HIV. Further, our intervention was delivered through the peak of the UK outbreak of coronavirus disease 2019 (COVID-19), part of the worldwide pandemic caused by severe acute respiratory syndrome coronavirus 2 (SARS-CoV-2). This is highly relevant since as a result of COVID-19, sexual health clinics in the UK had to significantly reduce their face-to-face appointments in order to reduce risk of infection. At the time of the study, services and health promotion initiatives needed to be delivered in a socially distanced way and remote text messaging for HIV prevention is one such approach.

The aim of the study was to develop and test a HIV prevention text messaging intervention targeting employees in the construction industry and delivered during the COVID-19 pandemic. The objectives were: (i) to develop a theoretically informed intervention using a participatory approach, behavioural mapping and fidelity testing; (ii) to assess the uptake, reach, engagement with and perceptions of messages; (iii) to consider the value of this technology-based approach to HIV-prevention for the target audience, and during the COVID-19 pandemic.

## 2. Methods

The study had approval from an institutional Research Ethics Committee as part of a wider research programme (Ref: FMHS LT12042016, amend no 2: 05112018).

### 2.1. Design

This study reports on the development, testing and outcomes of a 10-week text messaging health promotion intervention aimed at increasing HIV awareness and promoting HIV testing in a cohort of 291 construction workers in the UK. Message development was based on a participatory design and included a focus group (*n* = 9) with message fidelity testing (*n* = 9).

### 2.2. Setting and Participants

The workplaces were 10 organisations in the construction industry (nine large, one small-to-medium enterprises (SMEs)) with sites in the East Midlands in the UK, hosting a total of 4649 employees. Participants in the study were workers (including employees, contractors and agency staff) who, during the recruitment period, were working at any of 16 sites owned by the participating organisations and had attended a health promotion event on the day they were recruited (*n* = 464). Participants are referred to as employees herein. Employees were eligible to take part in the text messaging study if they owned a mobile phone, their comprehension of the English language was sufficient to read and understand the participant information and messaging, and they were present at one of the host sites on the day they were approached.

### 2.3. Procedure

Recruitment to the study took place at one of 21 workplace health check events between August 2019 to February 2020 at the participating organisations. The health check events were delivered as part of a wider research programme called Test@Work; further details on site recruitment, health checks and study findings are therefore reported elsewhere [53]. Here, the events were simply used as a platform to access employees in the construction industry and recruit employees into ‘Test@Work Texts’ (a follow-on workplace text messaging study conducted during the COVID-19 pandemic). From the employees who attended a workplace health check event (*n* = 464) those who were eligible (*n* = 426) were invited to take part in Test@Work Texts. All of these participants had taken part in an individual health check on the same day, which included their choice from a range of optional health assessments (weight, height, body mass index (BMI), waist-hip ratio, blood pressure, mental wellbeing and a rapid point of care HIV test). All participants had therefore received individualised health check results with tailored advice that had been provided by a member of a health promotion team. They had also received a take-home information pack with generic health guidance around healthy eating, physical activity, diabetes risk and heart health, musculoskeletal health, and mental wellbeing. For Test@Work Texts, interested employees provided a mobile phone number to a project researcher and were informed that they would receive a series of health promoting messages starting approximately four weeks later. To maximise response rate in this ‘real-world’ research [54], at the point of recruitment, participants were able to self-select to enter a prize draw to win £100 of high street shopping vouchers. Employees were not required to provide a reason for declining to take part, although if they stated a reason, it was recorded. Duplicate and incorrect numbers (i.e., one digit missing) were omitted. Since employees were recruited from a workplace health check event which included the option for them to take a rapid HIV test the same day, data were collected for self-reported prior testing history (before the event) as well as the number that opted in for the workplace HIV testing (at the event from which they were recruited). Section 3.1 outlines the data that were collected relating to delivery and engagement with the text messaging programme. The participants were sent messages over a 10-week period between March and June 2020. The intervention therefore occurred through the initial peak of COVID-19 in the UK, which was declared a pandemic by the World Health Organization in February 2020, with a nationwide lock down occurring in March 2020.

### 2.4. ‘Test@Work Texts’ Intervention

#### 2.4.1. Intervention Description

The text messaging intervention included a series of 29 messages delivered over 10 weeks, of which 23 messages promoted health (one-way push messages), two were welcome and close messages, and four invited feedback replies (two-way pull messages). Inclusion of interactivity has been shown to enhance engagement with text interventions [55,56]. Opt-out instructions were provided in the welcome message. Messages were designed to be 160 characters, the length of a standard text message. The focus of the messaging intervention was to promote HIV awareness and testing. Of the 23 health messages, 12 promoted HIV awareness and testing whereas 11 promoted other areas of health (e.g., physical activity and diet, musculoskeletal health and stress). The general health topics aligned with those included in a previously developed toolkit for employers on workplace health promotion and HIV testing [57]. The decision to promote HIV awareness and testing alongside other aspects of health was based on prior participatory research showing that participants prefer this approach since it helps to normalise HIV and testing [52] due to perceived stigma around HIV that persists both in society and the construction industry [43,58,59]. Nineteen of the messages contained web links to evidence-based resources related to the message content since information coming from credible sources can promote positive behaviour change [60]. These web links directed to information that was freely accessible to the general public (e.g., Public Health England, NHS England, SH24.org). Message creation, mapping to behavioural models and message validation processes followed the recommended steps for developing and pre-testing a text messaging programme for health behaviour change set out by Abroms et al. [18]. Development and reporting was guided by the mobile health (mHealth) evidence reporting and assessment (mERA) checklist [61] and the template for intervention description and replication (TIDieR) [62] (available in Appendix A).

#### 2.4.2. Message Creation and Behavioural Mapping

Message development was a participatory process, overseen by a medical trainee (MM) and health psychologist (HB). The message set was co-created in a workshop (*n* = 9) consisting of two HIV nurses, two medical trainees and five lay people (aged 25–60 years). Of the lay people, three were members of the public with no prior exposure to sexual health services or HIV interventions; the other two worked as receptionists at a sexual health clinic and therefore had some prior exposure to sexual health services and associated client groups.

HIV-related messages were designed to promote HIV awareness and reduce barriers to HIV testing. Barriers to HIV testing, HIV prevention and risk behaviours were informed by the literature and prior public engagement work, and mapped to elements of the COM-B model [63] (Table 1). The COM-B model is used extensively in behaviour change interventions, and it assumes that behaviour is the result of an interaction between three components: capability, opportunity, and motivation. The model has been used to identify enablers and barriers to sexual health services and workplace health interventions in several studies [60,64,65,66,67,68,69]. Messages were designed to include the common elements of effective behaviour change techniques for HIV and STI health promotion as identified in a prior systematic review: (a) problem solving; (b) feedback on behaviour; (c) social support (unspecified); (d) instructions on how to perform the behaviour; (e) information about health consequences; (f) information about social and environmental consequences; (g) demonstration of the behaviour; and (h) credible source [60].

#### 2.4.3. Message Validation: Fidelity Testing

Message validation was completed in a focus group (*n* = 9) included the participants described in 2.4.2. The aim of the focus group was to test the message content and language used for the health communications. This was to identify any additional HIV-related messages that had not arisen from the literature and to ensure the language of messaging both communicated the intended health message and was appropriate for working-age adults. Following a fidelity testing process previously used for text messaging content development [18,70] participants were asked to answer five questions with additional opportunity for providing free text responses: (a) What do you think are the advantages of getting a regular HIV test?; (b) What do you think are the barriers that prevent people from getting regular HIV tests?; (c) What do you think could help people talk more about HIV and HIV testing?; (d) Who do you think is most affected by HIV issues?; (e) Is there a message regarding HIV that you think is important to convey to people?.

To ensure messages were clear and conveyed the intended information or instruction, participants were then asked to interrogate and discuss each proposed message with the following questions: (a) What is this message saying?; (b) What do you think the message intends to mean?; (c) Who do you think this message is targeting?; (d) How do you feel about this message?; (e) What is the message telling you to do?.

Finally in order to identify strong messages and prevent repetition, participants were asked for all proposed messages to indicate: (a) What is the strongest message?; (b) What is the weakest message?; (c) Which message did you find most novel (original, unique, surprising)?; (d) Is there any message that you had already heard about before?; (e) Are there any messages you feel say the same thing or are unnecessary?.

A total of 34 health-related question items were interrogated. Based on user feedback, texts were edited for clarity and the total message set was reduced to 23 items, 12 relating to sexual health and HIV and 11 for general health topics. The complete message set mapped to relevant COM-B constructs is available in Appendix A. The medical trainee and one lay person from the focus group then received test texts of the final message set to check for readability and correct any issues with the functioning of web links, one using an android phone and one an Apple iPhone, the two most common handsets on the market [12].

#### 2.4.4. Message Delivery and Tracking

Texts were sent using textlocal.com, a UK-based messaging service that allowed pre-defined messages to be sent at scheduled intervals. This service has been used in previous text messaging health promotion interventions [17]. Texts were sent at regular intervals following a set schedule with messages delivered as two to three texts per week, on regular days of the week and times. This is because regular rather than ad-hoc messaging has been shown to enhance intervention engagement [55]. Texts were sent Tuesdays at 10.30 a.m. (a common break time for construction workers), Thursday evening at 7 p.m. (to reinforce safe sex information before any sexual encounters that may occur on Friday evenings), and Sundays at 5 p.m. (when most construction industry workers would not be working and any unsafe sex practices that had occurred over the weekend would be fresh in their memory). When only two messages were to be sent in a week then the Thursday text was omitted. The distribution of messages across weeks can be found in Appendix A. Participants in the focus group showed no preference for message frequency or delivery times and so the distribution pattern was informed by a previous study [52] and discussion with two construction workers who were not participants in the study. Texts were sent between March and June 2020 in two separate groups, each group starting their 10-week period on a different date according to when they were recruited at a workplace health event. However, each participant within a ‘group’ had the same start and end date and the intervention was therefore run as a cohort on two occasions with both cohorts completing the whole intervention during the COVID-19 pandemic. Both schedules followed the same pattern for each group. Texts were sent alternating between a HIV-focused message and a generic health message which reflected the overall health promotion goals of the project as described above. Message validity was 24 h. It was not possible to determine why some messages were not properly delivered (e.g., lack of signal, technical issue) and as a result any mobile phone number noted to have not ‘successfully received’ the message within the 24 h was simply sent the next message in sequence. Clicks to short links within texts were also tracked. In the final four interactive messages, replies were stored by the automated service in an inbox by telephone number and then anonymised at the end of the intervention.

#### 2.4.5. Automated Data Collection

The messaging service allowed the collection of message statistics such as: the number of messages sent, number of messages successfully delivered, number of messages read, and the number of people who opted out of the messaging protocol. At the end of the text messaging period, one hour after receiving the final information text, participants were sent a message of thanks for their participation followed by an instruction that they would receive four messages in succession to which they were asked to respond by text reply (at normal message rate). These four data collection messages were delivered at spaced intervals of 15 min to allow time for replies. They invited participants to indicate whether they had undertaken a HIV test since the workplace health check event (Q1: yes/no), whether they found the text messages useful (Q2: yes/no), whether they learned anything new from the messages (Q3: yes/no), and whether they would make any changes or take actions to look after their health as a result of the messages (Q4: free text replies). These items were successfully used for data collection in a previous text messaging health promotion intervention delivered to via the workplace setting [52]. For Q4, replies were assigned a value by the project team on whether the response was positive or negative in tone.

#### 2.4.6. Assessment of Feasibility, Reach, Acceptability and Engagement

Feasibility of text messaging as a mode of intervention delivery was determined by a participation response rate of >65% (e.g., % uptake of those who were invited). Reach of the intervention was defined as the percentage of messages successfully delivered, with success pre-defined as >75%. Acceptability of the intervention was determined by the number of participants opting out of the messaging, with success pre-defined as <20% opt-outs. Engagement was determined by the number of messages read (as a % of those delivered) with success pre-defined as >75% (e.g., engagement with the intervention); and by the number of responses to each of four data collection measures (e.g., engagement with two-way messages as a mode of data collection) with success pre-defined as >60% of participants responding to at least one item. Pre-determined rates were established based on rates used in fidelity testing of other digital interventions [57], adapted with team consideration of intervention delivery occurring during a global pandemic.

#### 2.4.7. Statistical Analysis

SPSS Statistics for Macintosh version 26 (2019) by IBM Corp, Armonk, NY, USA was used to perform statistical analysis. Pearson correlation was used to compare message number and number of delivered messages as a percentage of sent messages. HIV and lifestyle web links were analysed using independent samples t-test.

## 3. Results

### 3.1. Participant Characteristics and Testing History

Of the 464 event attendees, 426 (92%) owned a mobile phone and were invited to take part in Test@Work Texts. Of these, 291 employees (age 17–70 years, 81.8%M, 18.2%F) consented to take part (68.3%), and 135 declined (31.7%). Employees were not required to give a reason for declining, but those that did reported being too busy, not knowing their own mobile number, not being interested in being contacted at a later date about their health, or there were language barriers that would prevent engagement with the messaging content. Some male employees declined due to concerns about their partners or families becoming aware that they had received messages about HIV and/or sexual health. Participant characteristics are shown in Table 2. With regards to testing history (prior to the event at which they were recruited), 23.4% (*n* = 68: 72%M (*n* = 49); 28%F (*n* = 19)) reported having had a HIV test previously. Over three-quarters of the sample (75.6%) had never tested prior to the event (*n* = 220: 85%M (*n* = 187); 15%F (*n* = 33)). Two participants (0.7%) declined to answer. A high proportion of Test@Work Texts participants had opted-in for a rapid point of care HIV test at the event at which they were recruited. Of the 291 participants, 245 (84.2%: *n* = 199, 81.2%M; *n* = 46, 18.8%F) had taken a rapid HIV test the same day they were recruited. Only 46 (15.8%: *n* = 39, 84.8%M; *n* = 7, 15.2%F) had opted out of the rapid HIV test at the workplace health check event.

### 3.2. Intervention Uptake, Reach and Acceptability

Findings are presented for uptake, reach and acceptability, benchmarked to the pre-defined criteria.

With regards uptake, the recruitment response rate was 68.3% (291 consenting out of 426 invited) which exceeds the pre-defined success criteria for uptake of 65%. In terms of reach, out of a total of 8439 possible messages (29 messages per 291 participants), over the course of the study a total of 7726 messages were successfully sent (91.6%), and of these 7020 were successfully delivered (90.9%). This exceeds the pre-defined success criteria for reach of 75%. Of the messages that were not successfully delivered, an average of 24 messages were undelivered per scheduled message (messages undelivered: min 20, max 38).

With regards to intervention acceptability, there were just 36 opt-outs overall (12.4% of participants); this is well below the pre-defined rate of <20% opt-outs. Opt-outs always occurred on days that messages were sent—they occurred immediately after the introductory message (*n* = 8), after general health and lifestyle texts (*n* = 7) or after texts relating to sex, drugs or HIV topics (*n* = 21) of which (*n* = 8) opt-outs occurred on or after the final message prior to the data collection items. 100% of people opting out declared themselves heterosexual, 8.3% of opt-outs were non-white British, and all but one (97.2%) spoke English as their first language. The age range of those opting out was 19–64 years, and the mean age was 37 years.

### 3.3. Intervention Engagement and Evaluation

100% of messages successfully delivered were then read, which exceeds the pre-determined success rate of 75%. Engagement and evaluation data are presented with relation to the number of successful delivered messages rather than the number of study participants. Engagement with individual messages is provided in Table 3 This includes the weekly breakdown of intervention messages sent and delivered, including opt outs, over the 10-week messaging period; numbers exclude the data collection follow-up messages.

Of the 24 messages, 19 included short web links to health-related information of which eight were HIV-related messages and 11 were related to other areas of health. Across all 19 messages that included links, the mean number of clicks per link was 35.2 (SD = 10.9; *n* = 669 clicks; range = 21–59). As a percentage of clicks per messages successfully delivered, the mean was 14.4% (SD = 4.15 clicks per message, with a range of 8.8% to 24.1%). For HIV-related short links, the mean number of clicks per link was 34.6 (SD = 13.6, *n* = 255 clicks; range: 21–59). As a percentage of clicks per messages successfully delivered, the mean was 14.2% of delivered messages, with a range of 8.8% to 24.1%. There was no significant difference in the number of clicks between HIV-related short links, and other areas of health (t = −0.195, *p* = 0.85). The most highly accessed link within the whole message set related to HIV home testing (message 11) which generated 59 impressions (24.1% of delivered messages)—almost one quarter of the participants independently accessed the link to gain information about how to access a home test kit. The least accessed link (message 24) generated 22 impressions (9.4% of delivered messages) and related to drug use and addiction.

Eleven of the messages included links relating general health and lifestyle information. For these 11 messages, the mean number of clicks per link was 35.6 (SD = 9.1 *n* = 414 clicks, range: 22–53). This accounted for an average of 14.5% of delivered messages with a range of 9.7% to 20.2%. The most highly accessed link from general health messages related to risk factors in diabetes (message 3) with 53 clicks (20.2% of delivered messages)—more than one fifth of the participants accessed information about diabetes risk factors. The least accessed link related to physical activity ‘moving more’, with 23 impressions (9.7% of delivered messages). There was a negative correlation been message number and clicks per message (*r* = −6.24, *p* = 0.01), suggesting that the number of times that short links were accessed reduced over the duration of the 10 weeks. The links included were valid at the time of publication, and access rates for each link are provided in Appendix A.

Of the 291 participants, 13.7% (*n* = 40) responded to at least one of the four data collection measures which is lower than pre-defined success criteria of >60% of participants responding to at least one item from the four two-way messages. Response to individual data collection items is shown in Table 4.

There was a 6.6% decline between those who completed the first and the last data collection question. Eight participants (21.6% of those responding to Q1) reported having taken a HIV test since the workplace health check event; these were employees that had taken an optional HIV test already, as part of that event, and therefore reported independently having had a repeat test following the messaging. Of respondents, 32% (*n* = 8) reported finding the messages useful (Q2), and 45.5% (*n* = 10) felt they had learned something new from the messages (Q3). Overall, most of those who responded to Q4 self-reported positive lifestyle changes to their diet, weight and exercise habits, alcohol intake and rest habits with relation to stress reduction. One participant who responded with ‘no’ indicated that this was because they already had a healthy lifestyle. There were very few negative comments about the messaging process, or the content of the messages. Just one participant perceived that the messages were repetitive, and another felt that some of the messages did not relate to them (specifically, drug use), although the same participant also referred to the messages being ‘informative’ and confirmed that they had made changes to their lifestyle as a result of the intervention. Only two participants made comments relating to HIV awareness and testing specifically; one participant reported that they would have taken a HIV test following the messaging if they had not already had one recently (at the recruitment event) suggesting potential for positive influence on intentions to test. Another participant raised a concern about HIV being included in the message content: *“…my biggest issue was you sending txts with HIV test written all over it, when my wife constantly saw these it caused issues”.*

## 4. Discussion

To our knowledge, this is the first study to develop and test a HIV prevention text messaging intervention targeting employees in the construction industry. Our study showed that, theoretically, informed text messages including HIV awareness and education alongside general health promotion is highly acceptable for adults in the UK construction industry. Delivered during the global COVID-19 pandemic, the intervention had high uptake (68.3% opt-in), wide reach, high levels of message engagement and low levels of attrition. Engagement with external web links accessed through text messages was high irrespective of health area, indicating that mobile messaging is a useful platform for HIV awareness and sexual health promotion, as well as other health and lifestyle behaviours.

In a similar sexual health study, recruitment rates were even higher (97%) [71]. Uptake in other text messaging studies has been highly variable, ranging from 6.7–81% [72,73,74,75,76,77]. This is likely due to the high degree of variability in study and intervention design, implementation, target audience and interactivity across studies. A difference between our study and many others is how our participants enrolled. In our study, despite the remote nature of the intervention, employees were initially approached and recruited face-to-face in their place of work (as opposed to a clinical setting), then provided with an opt-out option with the introductory message. This may have increased the uptake since other studies have required participants to actively send a text message to be enrolled [77], be enrolled at a focus group [78], or be enrolled by a healthcare practitioner [73].

Reach of our study was high, with the majority (91%) of intervention messages being successfully delivered, with 90.9% of total texts (including introductory messages, intervention and data collection messages) successfully delivered. We observed high retention of users in the study (87.6% fully completed the intervention, only 36 opt-outs). This is broadly comparable with other studies where retention rates range from 75% to 95% [13,75,76,79,80], with 90% retention of participants in a sexual health study [71].

We observed high engagement with the messaging intervention. Up to one quarter of message recipients actively clicked on external web links for sexual health services, and up to one fifth of recipients clicked on external web links to seek more information from a health and lifestyle text. This is a higher engagement rate than might be expected compared with text messaging advertising where click rates range from 6–19% [81]. Most notable was the engagement with the HIV-related web links compared with other health messages, which had the greatest clicks on short links per message (compared with other general health messages in the intervention). The highest number of clicks per message was observed for the HIV home testing details, and for free condom services, demonstrating that the intervention was successful in engaging construction employees in HIV awareness and prevention education. HIV postal testing kits are becoming increasingly popular in the UK, with 138,453 home testing kits ordered through the eSexual Health service alone in 2018 [4]. Self-empowerment, privacy, confidentiality, convenience and opportunity to test have all been cited as preferences to test at home [82,83,84] and is supported by the high level of interest in this text message web link. The unavailability of standard sexual health services during the COVID-19 pandemic may have contributed to the higher level of interest in this particular message, with individuals needing to seek alternative means of accessing testing and other services.

Regarding engagement with external web links, it is difficult to make direct comparisons with other studies as few have reported the use of web links, and studies rarely measure engagement with text messages objectively, using tracking data. For example, one study reported engagement with external web links to be 56% which is higher than we observed [75], although web links were included in only 33% of messages (compared with 83% in our study) and their finding was based on subjective self-reports rather than an objective measure of message engagement. When compared to objective measures, we found satisfactory levels of engagement; in general SMS advertising, click through rates are estimated to be much lower [81].

Text messaging studies with high engagement often required acknowledgment from users of receipt of the text or group of texts, albeit for interventions aimed at other conditions such as type 1 diabetes rather than sexual health [24,74]. This was not required in our intervention, which was therefore very low cost. Inserting interactivity such as testing knowledge from true/false questions via bidirectional messages has shown to maintain interest and enjoyment [75], but would require increased manpower and developmental cost (as these schemes often require monitoring or custom software). The cost of sending two-way text messages message is also 50% higher than a standard message, yet may still provide a lower cost option than face to face intervention [27]. Cost versus benefit of these approaches could be explored in future.

We used four two-way messages to collect data at the end of the messaging intervention. Despite our high engagement and retention rates, response to the interactive data collection messages was low (13.7% of respondents replying to at least one data collection text). Low responses have also been reported in two-way text messaging surveys regarding sexual health and contraception, where response rates have been reported to be 40–44% [13,77,85] and even as low as 5% [76]. A rapid review by Pedder et al (2020) indicated that the response rate of text messages is highly variable and commonly decreases over time (with the exception of those involving pregnant women or type 1 diabetes) [86]. A sexual health survey with higher response rates than seen in most studies showed 59% of participants completing the first of three questionnaires and 39% completing the final one, though these rates are not directly comparable since the questionnaire was made available online to complete later and remained available for six weeks [71]. It is also feasible that heightened stress and anxiety seen in lockdown scenarios due to COVID-19 [87] may have distracted focus from these health promotion texts, since some recipients may have felt that replying was not timely or relevant under unprecedented global circumstances.

In our study there was a slight decline (6.6%) in the number of participants who completed the first and the last data collection question. This has been found elsewhere; several studies have shown that participants who begin surveys by text or email drop by between 18–20% between the first and last questions. If we compare this with telephone only questionnaires, 100% of questions are usually answered [88]. It may be that participants were experiencing fatigue with texts by the end of our intervention and the start of the text-based data collection. Fatigue in receiving texts has been observed in other text intervention studies, e.g., in one study promoting physical activity, where participants were required to reply to 5 daily texts over 4 months there was a decrease in response times as the study progressed, especially at weekends [89]. Financial incentives have been shown to improve response rates [75,90]. However, one of the advantages of SMS interventions is their low implementation cost so this type of incentivisation may be prohibitive. Future studies could create an online form to complete as part of the survey rather than bidirectional texts, allowing a greater number of questions and data to be collected, and allowing people to respond in their own time and reduce text messaging cost. Our findings suggest that while the text messaging intervention was very well received and had high engagement, alternative approaches to data collection may be required. It is not clear whether the challenge in our sample was the use of text messages to collect data (e.g., the platform for data collection), or whether the barrier was the focus of the data collection messages (e.g., a focus on HIV and sexual health). It is possible that response may increase with greater interaction, on-demand information and variation in delivery, through algorithmic approaches to text message replies, custom SMS gateways or peer led two-way SMS services although these approaches would occur an increased manpower and/or software development cost, contrary to the low cost appeal of SMS interventions [27,91].

Of those who responded to question 1 (*n* = 37), 21.6% (*n* = 8) reported having an HIV test since their HIV workplace test. This figure is promising, considering these participants had already taken an HIV test at the workplace health promotion event at which they were recruited, and these participants are therefore reporting repeat testing. This suggests that our intervention may help to increase the number of individuals who meet UK recommendations for HIV testing since it is currently recommended that those who are sexually active are tested for HIV once every three months and a minimum of once per year [92]. A 2019 report by Pubic Health England show that, repeat testing within a year was highest for gay men and black African heterosexuals (20.9% and 5.4% respectively) [9]. Data were not published for non-African heterosexual retesting, despite non-African heterosexual attendees and testing accounting for 73% of tests within a sexual health service in 2018 [4]. It is promising to see a proportion of our respondents repeat testing for HIV, especially heterosexual men who made up a large proportion of our study participants (96.6%).

Participants who responded to data collection texts were broadly positive and satisfied with message content. A small number of participants had privacy concerns (with regards messages being seen by partners) demonstrating the sensitive nature of some of the message content. This has been mirrored in other studies where participants felt more comfortable with health related texts compared to sexually themed messages [85] or were worried they may be embarrassed if someone saw the content of the messages [93]. A small number of participants felt that the messages were not appropriate to them—i.e., messages with a focus on HIV and drug taking. Our intervention included working age adults from a wide age range (younger and older workers), and this suggests there may be sub-groups within this construction sample for whom our message language and content may need refining. Even in studies of a single age group (e.g., adolescents) there are individual differences in language preferences, with some preferring casual language with others perceiving this informality as messages being dumbed down [94]. A comparison of SMS sexual health services in the U.S. showed that people value control over the language of text messages [93], although this level of fine tuning was not practical for the current study. There are also age-related differences in risk perception (irrespective of risk behaviours), since older audiences generally perceive HIV as a disease that affects gay men whilst still themselves taking part in sexual risk taking behaviours [84,95,96]. In our study, 72.9% of participants were working adults aged over 30 years, and the majority (97.3%) self-identified as heterosexual. Although content was developed through a participatory approach, it may be that the language and focus of some of the messages could be less appealing to an older workforce. A minority of respondents commented on the repetitive nature of texts. Other studies have reported that frustration can arise from the automatic nature of text programs including predictability of message delivery and repetitiveness [97]. Nevertheless, the significant engagement of our sample with short web links coupled with the low opt-out rates suggested that the messaging intervention on the whole had widespread appeal.

Forty-five per cent of the respondents to data collection messages stated that they had learnt something new from the text messages, and this aligns with previous studies in which participants admit that regular texts may teach them something they had not anticipated [93]. When compared to an Indonesian smoking cessation and sex education study which reported 95% of respondents learnt something new [85], our figure appears low. In studies targeting adolescents, 83% of participants reported learning new information [75]. However, a key difference between our study and many other SMS studies is that all of our participants had already undergone a face-to-face health check covering all health areas included in our messaging intervention, as well as face-to-face HIV testing and counselling. They had also been provided with an information pack including health materials covering both general and sexual health. The opportunities for learning ‘new’ information from the texts was therefore lower than might be expected if the intervention was delivered to employees who had not attended a health event. It is therefore encouraging that almost half the text recipients still reported having learned something new from the messages which indicates the value of this technology-based intervention. Indeed, a systematic review on STI prevention showed that face-to-face and technology-based interventions are both equally as effective at delivering health information [27]. Our participants reported making positive changes to health and lifestyle behaviours with relation to alcohol consumption, physical activity and diet, smoking, stress and sexual health.

### 4.1. Strengths, Limitations and Recommendations

These findings are important since the vast majority of sexual health promotion studies target adolescents or young adults (college students); this study therefore contributes to a small number of studies promoting sexual health in working-age adults. This is one of only a few studies that has mapped message content to a theoretical framework, with only 32% of interventions being theoretically informed [98]. Our study objectively measures engagement with SMS messaging through tracking of short links, whereas many studies rely on self-reports.

While our uptake was reasonably high, it should be noted that 31.7% of employees chose not to take part. Although all participants spoke English, a small number declined due to language barrier and therefore in future interventions, it may be useful to consider the possibility of study materials and messages being available in different languages, which was beyond the scope of this study. As such, there is a possibility that health promotion information may not have reached those with less health awareness, less knowledge or lower capacity to seek out health information, and this has been mirrored by participants in other programs [93]. For those that did not opt-in, or those who dropped out, offering an alternative messaging format such as email may help to increase retention [99]. We did not collect data on number of sexual partners and sexual habits prior to the study and so we are unable to comment of the effectiveness of these texts on risk behaviour (although this was not an objective of this study). Since our focus was on development, uptake, reach and engagement rather than outcomes, there was no control group, and we did not collect clinical follow-up data. We are therefore unable to determine the effectiveness of the intervention on HIV testing and this is beyond the scope of our study, although we do have self-reports of testing behaviour. We are not able to determine the reasons why some of the sent messages were not successfully delivered. In these instances, it is possible that participants’ phones were switched off or out of range for an extended period, the phone had the SMS capability switched off, the carrier was experiencing technical difficulties, or the number was roaming (abroad). However, the negative correlation between message number sent and percentage of messages successfully delivered, may suggest that some users were blocking message delivery without actively opting-out from the messages. Modern smart phones allow messages to be blocked or put into a ‘do not disturb’ list so they are documented as undeliverable but messaging software cannot distinguish this. Lower than expected response rates to the data collection texts, feedback about repetitiveness and decrease in successful message delivery and short link engagement over time all suggest that our 10-week intervention may have been too long for some users. While our engagement rates and retention were broadly comparable to other studies, future studies might consider delivering a smaller number of messages, including bidirectional texts at different points to test knowledge, customising content and including variable/on demand delivery schedules.

Although our SMS intervention was a success, technology is rapidly evolving and attitudes toward SMS messaging may be changing. Mobile phone usage has changed in the last decade, with average data usage climbing from 0.2 GB per month in 2012 to 1.9 GB per month in 2017 and continues to increase, while SMS usage has decreased with outgoing SMS messages reducing from 162 in 2012 to 82 in 2017 [12]. In 2013, several studies indicated younger audiences favoured SMS communication [100,101] but more recently younger audiences have shown to prefer mobile applications for education over SMS messages [102]. Apps have the ability to reach a wider audience, with an app for adolescent sexual education in New York being downloaded over 22,000 times [103]. A similar SMS study aimed at adolescents found that they sought personalisation of content and message timings [93] which could be achieved via electronic and app based approaches, which would allow information to be accessed on demand, while also allowing fine measurement of engagement, information requested and frequency which is not measurable in the same way with mobile messaging [104]. With a lack of a ‘one size fits all’ template for sexual health intervention, and few studies conducted with working-age adults, it would be beneficial to study the most appropriate mechanism for delivery of messaging to different age groups in the future.

### 4.2. Application of the COM-B Model

The COM-B is a very widely used model used to help identify what needs to be changed in order for an intervention to be effective. The model served as a useful guide to conceptualising how capability, opportunity and motivation could be targeted through our messaging intervention design. However, there is scope for further exploration of whether the COM-B is sufficiently attentive to diversity (e.g., in ethnicity and other demographics and employment circumstances) amongst our target audience, since the COM-B has been criticised for over-simplifying understandings of sources of behaviour and individual responses, with the potential to ignore variation in need [105]. Nevertheless, this model has been used previously in the area of sexual health and health testing uptake, for example, to explore barriers and facilitators to uptake of chlamydia testing [64] and to inform HIV prevention interventions, such as HIV self-testing programmes [106].

However, some challenges have been highlighted with relation to the application of COM-B as an overarching framework in the context of HIV testing. It has recently been proposed that a more nuanced framework of motivation may be required to examine the influence of social norms drawn from peers, community, and society and the impact of these norms on shaping engagement with HIV testing interventions [106]. In our context, anecdotal comments made by participants who took part in the health checks indicated that for many, participation was opportunistic (e.g., a health event taking place at their workplace), yet social norms within the construction industry appeared to play a key role in participants’ decisions around engagement in health behaviours more broadly (e.g., social norms, particularly among younger males, relating to alcohol or substance use, and risky sexual behaviours), as well as help-seeking behaviours (e.g., the influence of masculinity on openness about health concerns). These issues will be explored more fully in an upcoming analysis of qualitative interview data collected from workplace health check participants in the construction industry. However, the centrality of social norms in the uptake of precautionary behaviours has been demonstrated previously in HIV prevention research, and as proposed by Witzel et al., may not be fully accounted for in the COM-B model.

### 4.3. Impact of COVID-19

Our intervention was delivered through the peak of the UK outbreak of coronavirus disease 2019 (COVID-19), part of the worldwide pandemic of COVID-19 caused by severe acute respiratory syndrome coronavirus 2 (SARS-CoV-2). This is highly relevant since, as a result of COVID-19, sexual health clinics in the UK had to significantly reduce their face-to-face appointments in order to reduce risk of infection. At the time of the study, services and health promotion initiatives needed to be delivered in a socially distanced way and remote text messaging for HIV prevention is one such approach.

National lockdown restrictions affected most UK employees, including those in the construction industry. A large proportion of the UK construction workforce was either out of work or furloughed during this intervention period (furlough is where an employee or worker agrees with their employer to stop work temporarily but stay employed). With a lack of clarity as to how long restrictions would be in place, we decided to deliver the intervention as intended, and it therefore took place wholly through the pandemic.

It should be noted that our message delivery times were designed around anticipated behaviours, e.g., there is as much as a 5-fold increase in alcohol consumption and binge drinking from Thursday to Saturday night [107,108], which has been associated with increased sexual risk-taking behaviours including unprotected sex, and payment of sex workers [109,110]. Our messages were timed to highlight topics before these anticipated events and after to allow reflection and action. However, patterns of behaviour (including risky sexual behaviours) will certainly have been impacted by the lockdown and pandemic more broadly.

It remains to be seen how COVID-19 restrictions have affected the sexual landscape; early data suggests there has been increased contraception use and decreased pregnancy desires [111] and more frequent intercourse with regular partners [112], but there is little to no information relating to sexual behaviours involving external or multiple partners. In regards to HIV, reported use of pre-exposure prophylaxis (PreP) is lower [113] and over 53% of sexual health services closed in April 2020 [114] with HIV services showing between and 80–90% drop in provision [115]. Although the exact impact of the pandemic on our intervention is not clear, this may partially explain why, in our study, the two most clicked links relating to sexual health and HIV were for home HIV testing kits and free condom services. In regards to general health, 43% report that mental health had worsened since outbreak, and 35% report that general health had become worse since outbreak [116]. It will take time to understand how this global event has affected working practices, health risk behaviours and sexual habits, and this will undoubtedly influence the design of future studies in the area of HIV prevention. The timing of this intervention is critical since it is known that COVID-19 infection is likely to be worse in those with a ‘weakened immune system’. Although not all people with HIV are considered at increased risk, COVID-19 infection is likely to be worse in those individuals with a CD4 count of less than 200, those with a detectable viral load and those who are not on treatment [117], further demonstrating the importance of finding methods to promote HIV testing and reducing late diagnosis of HIV.

## 5. Conclusions

Mobile phone messaging for HIV awareness and prevention during the COVID-19 pandemic was well-received by working-age adults in the construction industry. The intervention had high uptake and engagement, and low levels of attrition. Employees actively engaged with messages, following signposting to general health, sexual health, and HIV-related websites and accessed details of home HIV testing services. After the intervention, the reported rate of repeat HIV testing was higher than the national average and participants reported making positive changes in health and lifestyle behaviours, including sexual health. In the context of COVID-19 restrictions affecting participants’ usual habits, text messaging was a valuable tool for delivering health-based information in the construction industry. However, alternative approaches to data collection should be investigated.

## Figures and Tables

**Table 1 ijerph-17-07819-t001:** Barriers to HIV testing and HIV prevention/risk behaviours mapped to the COM-B model.

**Capability**	**Psychological**	Lack of understanding about HIV and testing Fear of test result Lack of confidence going for a test Lack of knowledge of how to prevent infection Stigma prevents partner notification Lack assertiveness to insist on condom use Does not like condoms or problems using them Forgetfulness Lack of provider knowledge Not aware of who can have HIV and how it is transmitted Unaware of areas of high HIV prevalence Unaware of rapid tests Unaware of other STD risks—not just about pregnancy Unaware of asymptomatic period of HIV Unaware of testing frequency
	**Physical**	Difficulty accessing a testing centre Difficulty obtaining an appointment Lack of knowledge about where to get tested Condoms not available
**Opportunity**	**Physical**	No time to get tested Not registered with a doctor (general practitioner) Does not want to deal with a receptionist Cannot afford condoms Unable to travel to a testing centre
	**Social**	Does not want people to know they are getting tested Stigma—belief that HIV is for gay people Stigma—belief that STDs are for being unclean Stigma—related to carrying condoms Friends do not get tested and do not talk about it Fear about perceptions of others towards them Lack of confidence in using testing services Social norms to not use protection
**Motivation**	**Reflective**	Been told to get tested Belief that they are low or no risk for HIV Belief that partner is faithful (‘so I don’t have HIV’) Assessing risk of partner based on their appearance Lack of knowledge of how testing stops STDs and HIV Lack of awareness of causes and implications of HIV and AIDS Beliefs about financial cost of condoms (‘too expensive’) Unaware of asymptomatic nature of HIV Poor enjoyment of condoms (lack of sensation etc.) Embarrassment and shame Beliefs around HIV (‘death sentence’, ‘no treatment’)
	**Automatic**	Unprotected sex Never had a test Anxiety about result Intravenous drug use Fear (related to HIV and testing or consequences of test) Shame Social isolation Low self esteem Risky sexual behaviour (paying for sex, sex whilst intoxicated, Chemsex) Deliberation

**Table 2 ijerph-17-07819-t002:** Participant characteristics.

	Male *n* = 238 (81.8%)	Female *n* = 53 (18.2%)	Full Sample *n* = 291 (100%)
	*n* (%)	*n* (%)	*n* (%)
**Age category (years)**			
17–30	65 (27.3)	14 (26.4)	79 (27.1)
31–40	64 (26.9)	14 (26.4)	78 (26.8)
41–50	62 (26.0)	10 (18.9)	72 (24.7)
51–60	36 (15.1)	12 (22.6)	48 (16.5)
61—70	11 (4.6)	3 (5.7)	14 (4.8)
**English as first language**			
Yes	223 (93.7)	50 (94.3)	273 (93.8)
No	13 (5.4)	3 (5.7)	16 (5.5)
Not stated	2 (0.8)	0 (0.0)	2 (0.7)
**Ethnicity**			
**White**			
British	204 (85.7)	35 (66.0)	239 (82.1)
Irish	4 (1.7)	0 (0.0)	4 (1.4)
Any other white	6 (2.5)	3 (5.7)	9 (3.1)
background			
**Mixed**			
White and black-Caribbean	3 (1.3)	1 (1.9)	4 (1.4)
White and black African	0 (0.0)	0 (0.0)	0 (0.0)
White and Asian	0 (0.0)	1 (1.9)	1 (0.3)
Any other mixed	0 (0.0)	1 (1.9)	1 (0.3)
background			
**Asian or Asian British**			
Indian	13 (5.5)	4 (7.5)	17 (5.8)
Pakistani	1 (0.4)	0 (0.0)	1 (0.3)
Bangladeshi	1 (0.4)	0 (0.0)	1 (0.3)
Any other Asian	0 (0.0)	0 (0.0)	0 (0.0)
background			
**Black or Black British**			
Caribbean	4 (1.7)	5 (9.4)	9 (3.1)
African	0 (0.0)	1 (1.9)	1 (0.3)
Any other Black	0 (0.0)	0 (0.0)	0 (0.0)
background			
**Other Ethnic groups**			
Chinese	0 (0.0)	1 (1.9)	1 (0.3)
Any other ethnic group	0 (0.0)	0 (0.0)	0 (0.0)
**Not stated**	2 (0.8)	1 (1.9)	3 (1.0)
**Sexual orientation**			
Heterosexual	230 (96.6)	53 (100)	283 (97.3)
Homosexual	1 (0.4)	0 (0.0)	1 (0.3)
Other	1 (0.4)	0 (0.0)	1 (0.3)
Not stated	6 (2.5)	0 (0.0)	6 (2.1)

**Table 3 ijerph-17-07819-t003:** Information messages sent and delivered during the 10-week intervention period.

Week	Messages per Person	Total Sent	Total Delivered	% of Sent Texts Delivered	Opt-Outs	% of Delivered Messages Opted-Out
1	3	847	784	92.6	15	1.9%
2	3	818	753	92.1	2	0.3%
3	3	809	741	91.6	3	0.4%
4	2	534	490	91.8	0	0.0%
5	2	536	488	91.0	0	0.0%
6	3	801	730	91.1	0	0.0%
7	2	528	482	91.3	6	1.3%
8	2	523	475	90.8	2	0.4%
9	2	520	456	87.7	0	0.0%
10	2	520	469	90.2	8	1.7%
Average	2.4	643.6	586.8	91.0	6	0.6%

**Table 4 ijerph-17-07819-t004:** Final data collection message statistics.

Question	Yes *n* (%)	No *n* (%)	Total Replies (*n*)	Replies as % Delivered Messages (%)
Have you had an HIV test since your Test@Work day?	8 (21.6)	29 (78.4)	37	15.9
Have you found these texts useful?	8 (32)	17 (68)	25	10
Have you learned anything new from these texts?	10 (45.5)	12 (54.5)	22	9.7
Will you make any changes or take actions to look after your health as a result of these messages?	13 (61.9)	8 (38.1)	21	9.3

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
