# Peer review of "Test@Work Texts: Mobile Phone Messaging to Increase Awareness of HIV and HIV Testing in UK Construction Employees during the COVID-19 Pandemic"

_ijerph, 2020, doi:10.3390/ijerph17217819_

Round 1

Reviewer 1 Report

The study entitled “Test@Work Texts: Mobile phone messaging to increase awareness of HIV and HIV testing in UK construction employees during the COVID-19 pandemic” done by Dr. Holly Blake et al is interesting, but it needs to be addressed several issues as below: Major comments 1. The supplemental tables/figures should not be included in the main text, they should be placed in somewhere such as supplemental files with the link to the paper. 2. How did you select study subjects, especially for the criteria of fluent in English language? Because in this study understanding of the message is so important, and 16 (5.5)% participates in this study do not speak English as first language. What happens if you remove they from the data for analysis? 3. The way of presenting the data makes confusing for the readers, it is difficult to follow and get the main main messages from the study, I think authors should reorganize the data and paper to have the better version 4. The article is completely descriptive, without a statistically significant comparison, could it be a research paper? Perhaps turning it into another acceptable type of paper in this journal?

Author Response

Thank you for reviewing our article and your feedback, we have carefully considered each point please find below responses to your comments and the attached revised manuscript reflecting any changes.

 1. The supplemental tables/figures should not be included in the main text, they should be placed in somewhere such as supplemental files with the link to the paper.

The supplementary files were included by the editorial team for your ease of reference, they were submitted and will be available in separate files for the final publication, please see the revised manuscript without the supplementary material in the text.

2. How did you select study subjects, especially for the criteria of fluent in English language? Because in this study understanding of the message is so important, and 16 (5.5)% participates in this study do not speak English as first language. What happens if you remove they from the data for analysis?

Study subjects were chosen from in person health checks described in the paper in section 2.3. All participants in health checks were offered the opportunity to take part in a follow up texts message study (this study).

The wording has been changed to account for this.  All participants spoke English (even if it was not their first language). English language comprehension at baseline was sufficient to read and understand the participant information and sign-up sheet as well as the instructions from the health volunteers that were delivering the in-person health checks at the events at which our participants were recruited.  No users were excluded because of English not being their first language, however a small number did decline citing language barrier as a reason. Since the study was to ascertain the uptake and reach of the text message intervention, and the feasibility of this type of intervention within the construction industry and with a population including migrant workers, it would not be appropriate to remove participants from the analyses (this is not an outcome study). However, we have reflected on this potential issue in the discussion:

“Although all participants spoke English, a small number declined due to language barrier and therefore in future interventions, it may be useful to consider the possibility of study materials and messages being available in different languages, which was beyond the scope of this study”.

We have also clarified this by rephrasing the eligibility criteria: “their comprehension of the English language was sufficient to read and understand the participant information and messaging”.   

3. The way of presenting the data makes confusing for the readers, it is difficult to follow and get the main messages from the study, I think authors should reorganize the data and paper to have the better version

We agree that the inclusion of supplementary materials within the main body of the paper is likely to have made the results section harder to read.  We have removed these from the main paper and included them as separate supplementary files and this has improved the flow of results. Comments from the other two reviewers indicate that the results are well presented and so we hope this reorganisation of tables has addressed the issue raised here. We have added a new table to show the response rate for the 4 data collection measures.

4. The article is completely descriptive, without a statistically significant comparison, could it be a research paper? Perhaps turning it into another acceptable type of paper in this journal? 

Thank you for your comment. We believe this is a research paper and hope that it will be considered acceptable for the journal. Our aims demonstrate that we are interested in the development process, reach, uptake and engagement with a text messaging intervention and believe the analysis and results address these aims. Since this is not a study of effectiveness, we did not set out to compare outcomes. We have added a sentence to ensure that this is very clear in the interpretation of findings and discussion.

Best wishes

The authors

Reviewer 2 Report

Congratulations on designing an innovative health promotion activity during the pandemic, that focuses on people who are less likely to be included in sexual health promotion activities in the developed world. This paper was a pleasure to read. It was clear, logical and provided enough detail for others to be able to replicate this study. It is unfortunate that only 13.7% (n=40) of participants responded to at least one of the data collection measures although a high proportion of engagement (about a quarter of the participants) with website links shows that recipients read the text messages and engaged with them, so this lack of response to data collection measures is less important. 

There were a tiny number of typo's which I have listed below.

Other than that, congratulations on a really good, interesting and well written paper.

Line 128 - what does SME mean? Does it mean small to medium enterprise? If so, this is not clear in the way it is currently written.

Line 279 - delete the word 'during' from this line.

Line 401 - (21.6%) of those responding to Q1, should read (21.6% of those responding to Q1)

Author Response

Thank you for reviewing our article and your comments on our paper. Follows are the amendments made and they are reflected in the revised manuscript.

There were a tiny number of typo's which I have listed below. Other than that, congratulations on a really good, interesting and well written paper.

Thank you for this feedback. Typographical errors have been corrected.

Line 128 - what does SME mean? Does it mean small to medium enterprise? If so, this is not clear in the way it is currently written.

Yes, the SME should have been in brackets, this has been amended,

Line 279 - delete the word 'during' from this line.

Deleted

Line 401 - (21.6%) of those responding to Q1, should read (21.6% of those responding to Q1)

Amended

Best wishes

The authors

Reviewer 3 Report

This study reports on the development, testing and outcomes of a 10-week text messaging health promotion intervention aimed at increasing HIV awareness and promoting HIV testing in a cohort of 291 construction workers in the UK. Overall, I found the intervention well-designed and delivered. 

  1. It is excellent that the message was informed by COM-B model. Any reasons to choose this particular model? Any advantages and disadvantages by using this model in the present setting?
  2. As discussed in the Introduction, construction works may be at increased risk for HIV. May the author provide more details about their sexual behaviors or other related behaviors (e.g., substance use)?
  3. I am wondering why there was no control group, and how this might limit the interpretation of the current findings.

Author Response

Thank you for reviewing our article and your comments.  Please find below responses to your comments and our revised manuscript to reflect these changes.

Best wishes

The authors

It is excellent that the message was informed by COM-B model. Any reasons to choose this particular model? Any advantages and disadvantages by using this model in the present setting?

Thank you for this helpful comment. We have reflected on this and have added a section on this to our discussion.

“The COM-B is a very widely used model used to help identify what needs to be changed in order for an intervention to be effective. The model served as a useful guide to conceptualising how capability, opportunity and motivation could be targeted through our messaging intervention design. However, there is scope for further exploration of whether the COM-B is sufficiently attentive to diversity (e.g. in ethnicity and other demographics and employment circumstances) amongst our target audience, since the COM-B has been criticised for over-simplifying understandings of sources of behaviour and individual responses, with the potential to ignore variation in need (Ogden, 2016).

Nevertheless, this model has been used previously in the area of sexual health and health testing uptake. For example, to explore barriers and facilitators to uptake of chlamydia testing (McDonagh et al, 2018) and to inform HIV prevention interventions, such as HIV self-testing programmes (Witzel et al, 2020). However, some challenges have been highlighted with relation to the application of COM-B as an overarching framework in the context of HIV testing. It has recently been proposed that a more nuanced framework of motivation may be required to examine the influence of social norms drawn from peers, community, and society and the impact of these norms on shaping engagement with HIV testing interventions (Witzel et al, 2020). In our context, anecdotal comments made by participants who took part in the health checks indicated that for many, participation was opportunistic (e.g. a health event taking place at their workplace), yet social norms within the construction industry appeared to play a key role in participants’ decisions around engagement in health behaviours more broadly (e.g. social norms, particularly among younger males, relating to alcohol or substance use, and risky sexual behaviours), as well as help-seeking behaviours (e.g. the influence of masculinity on openness about health concerns). These issues will be explored more fully in an upcoming analysis of qualitative interview data collected from workplace health check participants in the construction industry. However, the centrality of social norms in the uptake of precautionary behaviours has been demonstrated previously in HIV prevention research, and as proposed by Witzel and colleagues, may not be fully accounted for in the COM-B model”.

References:

McDonagh LK, Saunders JM, Cassell J, Curtis T, Bastaki H, Hartney T, Rait G.  Application of the COM-B model to barriers and facilitators to chlamydia testing in general practice for young people and primary care practitioners: a systematic review. Implement Sci. 2018 Oct 22;13(1):130.  doi: 10.1186/s13012-018-0821-y.

Ogden, J. Celebrating variability and a call to limit systematisation: The example of the behaviour change technique taxonomy and the behaviour change wheel. Health Psychol. Rev. 2016, 10, 245–250

Witzel TC, Weatherburn P, Bourne A, Rodger AJ, Bonell C, Gafos M, Trevelion R, Speakman A, Lampe F, Ward D, Dunn DT, Gabriel MM, McCabe L, Harbottle J, Moraes YC, Michie S, Phillips AN, McCormack S, Burns FM. Exploring Mechanisms of Action: Using a Testing Typology to Understand Intervention Performance in an HIV Self-Testing RCT in England and Wales. Int J Environ Res Public Health 2020 Jan 10;17(2):466.  doi: 10.3390/ijerph17020466.

As discussed in the Introduction, construction works may be at increased risk for HIV. May the author provide more details about their sexual behaviors or other related behaviors (e.g., substance use)?

We have amended the text to provide more detail:

International studies have found a high prevalence of risky sexual behaviour amongst construction workers (44%) [27] (Kassa et al, 2013), such as sex with non-regular partners or while under the influence of alcohol, coupled with poor knowledge about sexual health and low rates of condom use [28] (Arora et al, 2014). Unhealthy lifestyle behaviours are common in the construction industry (e.g. poor diet, smoking, problem alcohol consumption, substance abuse) [26,29,30], with increased risk of mortality from drug use and accidental drug poisoning in construction workers (e.g. male painters and decorators, bricklayers and masons, plasterers, and roofers and glaziers) [31] (Coggon et al 2010).

References

Arora VK, Sharma S, Mahashabde S. Sexual behaviour among migrant construction workers in Indore. International Journal of Medical Science and Public Health, 2014; 3; 5; 574-577.

Coggon D, Harris EC, Brown T, Rice T, Palmer KT. Occupation and mortality related to alcohol, drugs and sexual habits. Occup Med (Lond). 2010 Aug;60(5):348-53.  doi: 10.1093/occmed/kqq040.  Epub 2010 Apr 20.

Kassa M, Tesfaye E, Alamrew Z. Risky Sexual Behaviour among Big Construction Enterprise Workers; Bahir Dar City, Amhara Regional State, Northwest Ethiopia. International Journal of Clinical Medicine, 2013, 4, 296-303 doi:10.4236/ijcm.2013.46052

I am wondering why there was no control group, and how this might limit the interpretation of the current findings.

Thank you for your comment. Since this is not an outcomes study, a control group was not required to meet our study aims which are focused on uptake, reach and engagement rather than outcomes.

“The aim of the study was to develop and test a HIV prevention text messaging intervention targeting employees in the construction industry and delivered during the COVID-19 pandemic. The objectives were: i) to develop a theoretically informed intervention using a participatory approach, behavioural mapping and fidelity testing; ii) assess the uptake, reach, engagement with and perceptions of messages; iii) to consider the value of this technology-based approach to HIV-prevention for the target audience, and during the COVID-19 pandemic”.

We would be unable to determine effectiveness of the intervention from this study, and it was not one of our aims to do so.

However, to ensure this is clear to the reader, we have added this to our discussion in the limitations section:

“Since our focus was on development, uptake, reach and engagement rather than outcomes, there was no control group, and we did not collect clinical follow-up data. We are therefore unable to determine the effectiveness of the intervention on HIV testing and this is beyond the scope of our study, although we do have self-reports of testing behaviour”.

Round 2

Reviewer 1 Report

The manuscript has been improved